# Stronger Than You Think:
# Benchmarking Weak Supervision on Realistic Tasks

**Tianyi Zhang**[1][*] **Linrong Cai**[2][*] **Jeffrey Li**[1] **Nicholas Roberts**[2] **Neel Guha**[3] **Frederic Sala**[2]

[1]University of Washington [2]University of Wisconsin-Madison [3]Stanford University

tzhang26@uw.edu, lcai54@wisc.edu

## Abstract

Weak supervision (WS) is a popular approach for label-efficient learning, leveraging diverse sources of noisy but inexpensive *weak labels* to automatically annotate training data. Despite its wide usage, WS and its practical value are challenging to benchmark due to the many knobs in its setup, including: data sources, labeling functions (LFs), aggregation techniques (called label models), and end model pipelines. Existing evaluation suites tend to be limited, focusing on particular components or specialized use cases. Moreover, they often involve simplistic benchmark tasks or de-facto LF sets that are suboptimally written, producing insights that may not generalize to real-world settings. We address these limitations by introducing a new benchmark, BOXWRENCH,[2] designed to more accurately reflect *real-world usages of WS*. This benchmark features tasks with (1) higher class cardinality and imbalance, (2) notable domain expertise requirements, and (3) opportunities to re-use LFs across parallel multilingual corpora. For all tasks, LFs are written using a careful procedure aimed at mimicking real-world settings. In contrast to existing WS benchmarks, we show that supervised learning requires substantial amounts (1000+) of labeled examples to match WS in many settings.

## 1 Introduction

Weak supervision (WS) aims to address the labeled data bottleneck for supervised machine learning. It uses multiple weak but inexpensive sources of signal and combines them into high-quality *pseudolabels* that can be used for training downstream models [34, 35, 39]. These weak sources can be diverse, including but not limited to: heuristic rules encoded into small programs, queries to knowledge bases, and pretrained models. Frameworks implementing WS are hugely popular and are widely applied in industry [3, 36] and academic settings [12, 42].

WS frameworks typically have a simple three-stage approach. First, they formalize weak sources into *labeling functions (LFs)*. In contrast to manual labeling, these can be automatically applied to an entire unlabeled dataset. Next, since LFs are inherently noisy and may conflict with one another, a *label model (LM)* is used to estimate the quality of each source (typically *without* accessing ground truth labels) and then to aggregate their outputs into high-quality pseudolabels. Finally, these pseudolabels can be used to train a downstream model. A vast literature studies variations on this basic recipe, with diverse approaches to crafting LFs, creating LMs, and noise-aware training of end-models [47].

For practitioners, a key question is ***when is WS useful?*** While it is natural to produce benchmarks that answer this, surprisingly, there has been relatively little work doing so. One reason for this may be the overall complexity of WS pipelines. The performance of a WS system varies with (1) the

---

[*]Equal contribution.

[2]The box wrench is the most ubiquitous and practical wrench.

38th Conference on Neural Information Processing Systems (NeurIPS 2024) Track on Datasets and Benchmarks.

underlying task and data, (2) the LFs, (3) the choice of LM, and (4) the choice of end model and training procedure. Several benchmarks predominantly focus on *only one of these*. For example, WRENCH [46] focuses primarily on evaluating (3), the LM, while AutoWS-Bench-101 [37] focuses on (2), the LFs, and specifically, techniques for automatically generating model-based LFs.

Recently, Zhu et al. [49] tackle the goal of quantifying the value of WS. They argue that the benefits of WS are often overestimated by showing that fine-tuning on only 50 ground-truth labels can achieve comparable—or better—results than certain WS approaches for many *benchmark datasets*. They suggest that WS may not be broadly useful, as obtaining 50 "clean" labels is rarely prohibitive, and data at this scale (or larger) may still be needed for tuning or evaluation even when using WS.

In this work, we show that these findings result *from the simplicity of existing datasets* rather than the inherent *weakness* of WS. In particular, we identify key issues with the current WS benchmarks that led to this result and show that WS may be ***stronger than is thought*** in more realistic settings:

1. **Benchmark datasets usually have too few classes, are balanced, or aren't specialized enough** to be representative of real-world datasets.
2. WS depends on the quality of LFs, and **LFs from current benchmarks can be improved**.
3. **Previous benchmarks do not capture the adaptability of LFs across task specification,** a key practical advantage of WS deployments compared to manual labeling.

We introduce a new benchmark, BOXWRENCH, that addresses these three challenges. It enables us to quantify the practical advantages of WS in a wide range of settings. Our findings indicate that even simple WS approaches often provide substantial value. We address the issues we identified by:

- Proposing new WS benchmarks based upon tasks that involve **high-cardinality label spaces, imbalanced classes, and/or require specific domain knowledge**.
- Showing that by adhering to careful LFs design practices, we can write effective LFs for these tasks that can even improve upon existing benchmark LFs.
- Using the MASSIVE dataset [10], we study simple but effective strategies for adapting existing LFs written for English data to parallel versions of the task in other languages.

Our benchmark consists of *five* text-classification WS tasks that showcase the power of WS in a variety of challenging real-world scenarios. For two of our tasks, we produce new LFs, while for one, we improve the existing LFs from WRENCH [46]. The design of these LFs follows a rigorous procedure that we release as part of our benchmark, acting as guidance for LF design and for WS benchmarking overall. We publicly release the code and other assets for our study at `https://github.com/jeffreywpli/stronger-than-you-think`.

## 2 Background and Related Work

We provide a brief background on WS techniques and benchmarking efforts. We note that the term WS can be overloaded, as it is also applied to other families of techniques that generally aim to learn from indirect or noisy forms of supervision [32], particularly in computer vision [28, 20]. Here, we use WS to refer to approaches that fall under *programmatic WS* [34, 47].

**Weak Supervision.** WS aggregates multiple imperfect label sources, each formalized as a *labeling function* (LF), to synthesize labels for unlabeled data. Perhaps the most popular types of LFs are heuristics or rules obtained from subject matter experts [33] encoded into programs. Other potential sources include knowledge-base queries, pretrained models, and more [16, 21, 8, 29, 17]. Many works in WS focus on improving either how LFs are crafted or how they are aggregated. For LF construction, variations such as learning small models on tiny amounts of data [43], or using code-generating large language models to craft LFs have been proposed [19, 18, 15]. For aggregation, the simplest technique is to perform a *majority vote* while other methods seek to infer (without using ground truth labels) the accuracy of the LFs with a *label model* (LM) and thus perform a higher-quality aggregation [33, 13, 40]. Orthogonal to both directions, other works study more effective strategies for training end models on WS-generated pseudo-labels [45, 24].

**WS Benchmarks.** Existing WS benchmarks typically focus on a particular component of a WS system or a particular use case. WRENCH [46] primarily benchmarks different label models and is therefore aimed at aggregation. AutoWS-Bench-101 [37], in contrast, studies the effectiveness of automated LF construction techniques. Finally, WALNUT [48] studies WS techniques in the context

of natural language understanding. All of these benchmarks are highly useful, but do not attempt to measure the value of WS techniques more broadly. A recent effort by Zhu et al. [49] tackles this question and finds that in particular settings, WS may not be of great value. Specifically, it suggests that only a small amount of labeled data is sufficient to train a supervised model to a level of quality equivalent to that provided by WS. We are inspired by this work, studying whether we can obtain similar findings across a broader range of realistic scenarios.

## 3 Methodology and Datasets

We establish the goals, problem setting, datasets, and experimental setup used in BOXWRENCH.

### 3.1 Goals

The ultimate goal of BOXWRENCH is to bridge WS research and practice by introducing more realistic benchmarks for WS. The first step towards such a goal is to gain a better understanding of the question: ***when is WS useful?*** To do so, we first gather a suite of datasets that addresses two key areas in which current WS benchmarks fall short.

1. Benchmark datasets tend to be simplistic, exhibiting properties not representative of many real-world problems. This includes having a small label space (often binary), balanced label distributions, and relying on general rather than domain-specific knowledge.

2. Current WS benchmarks are used with de-facto LF sets that vary in quality. A poorly written LF set may also result in a less realistic benchmark (e.g. if a task involves domain expertise but experts were not involved in writing the LFs).

To address the first issue, we introduce WS tasks that directly target the aforementioned gaps: focusing on those with greater class counts, class imbalance, and domain-specificity. To address the second issue, we place care into writing higher-quality LFs for all datasets, including improving existing LFs. Using these datasets, we aim to measure *how many labeled examples are needed* before supervised learning catches up to WS techniques. We formalize this notion by plotting the performance curves of WS techniques and fully supervised learning (as functions of the number of labels) and analyzing the **crossover points** where these curves intersect. To show that WS is effective, the crossover point in which supervised learning surpasses WS should be sufficiently high, i.e., WS cannot be matched by simply labeling a trivial amount of examples. Using crossover points to measure the effectiveness of WS, we aim to establish a regime in which WS is practically useful on our suite of more challenging and realistic datasets.

### 3.2 Problem Formulation

Let $\mathcal{D} \subseteq \mathcal{X} \times \mathcal{Y}$ be our data distribution. We first sample an unlabeled training set with $n_{\text{train}}$ examples $X_{\text{train}} = [x_i]_{i=1}^{n_{\text{train}}}$, and a small labeled validation set of $n_{\text{val}}$ examples: $D_{\text{val}} = [X_{\text{val}}, Y_{\text{val}}]$ with $X_{\text{val}} = [x_i]_{i=n_{\text{train}}+1}^{n_{\text{train}}+n_{\text{val}}}$ and $Y_{\text{val}} = [y_i]_{i=n_{\text{train}}+1}^{n_{\text{train}}+n_{\text{val}}}$ with $x_i \in \mathcal{X}$ and $y_i \in \mathcal{Y}$. We are interested in learning a function $f_\theta : \mathcal{X} \to \mathcal{Y}$ that minimizes the expected risk $R(f_\theta) = \mathbb{E}_{(x,y) \sim \mathcal{D}}[L(f_\theta(x), y)]$ where $L$ is a loss function and $\theta$ are the model parameters. To estimate $R(f_\theta)$, we assume the existence of an i.i.d. test set $D_{\text{test}}$, for which labels are also available. In the case of WS, we assume that labels for $X_{\text{train}}$ are provided by a set of labeling functions $\text{LF}_{\text{ws}} = \{\lambda_j\}_{j=1}^m$ where each LF, $\lambda_j : X \to Y \cup \{-1\}$ encodes some heuristic that labels or abstains (denoted by -1) on each input $x_i$. Using a label model LM that aggregates the LF votes, $\lambda_{i,j} = \lambda_j(x_i)$, we obtain $\widehat{y}_i = \text{LM}\left([\lambda_{i,j}]_{j=1}^m\right)$, namely, the weak label for $x_i$. We construct the weak labels for the training set as $\widehat{Y}_{\text{train}} = [\widehat{y}_i]_{i=1}^{n_{\text{train}}}$. We then use $D_{\text{WS}} = [X_{\text{train}}, \widehat{Y}_{\text{train}}]$ as the weakly labeled training set, to train a model $f_{\theta_{\text{ws}}}$.

Following Zhu et al. [49], we compare the performance $f_{\theta_{\text{ws}}}$ with two other models that train directly on the validation labels: $f_{\theta_{\text{SUP}}}$ is the model trained only on $D_{\text{val}}$, which serves as a critical baseline for assessing the usefulness of WS. $f_{\theta_{\text{CFT}}}$ is a model that results from using $f_{\theta_{\text{ws}}}$ as an initialization and further performs continuous fine-tuning on $D_{\text{val}}$. Weak supervision is considered useful given $D_{\text{val}}$ if the performance of either $f_{\theta_{\text{ws}}}$ or $f_{\theta_{\text{CFT}}}$ remains significantly higher than that of $f_{\theta_{\text{SUP}}}$.

Table 1: The datasets used in BoxWRENCH and their metadata. For MASSIVE, the dataset sizes are the amounts *per available language*.

| Dataset | Class | Train | Valid | Test |
|---|---|---|---|---|
| Banking77 | 77 | 9,003 | 1,000 | 3,080 |
| ChemProt | 10 | 12,600 | 1,607 | 1,607 |
| Claude9 | 9 | 5,469 | 200 | 2057 |
| MASSIVE{18, 60} | 18, 60 | 11,564 | 3,305 | 1,651 |
| Amazon31 | 31 | 131,781 | 5,805 | 17,402 |

## 3.3 Datasets

Existing WS benchmarks often exhibit low class cardinality (under 5), highly balanced label distributions (i.e. near uniform), and simplistic underlying tasks that do not require domain-specific knowledge. However, real-world tasks often do not conform to these assumptions. Thus, to evaluate WS in realistic regimes, we propose a collection of datasets[3] with varying levels of class cardinality and imbalance, as well as requirements for domain expertise. We describe each of the datasets used in BoxWRENCH, with their metadata shown in Table 1, and describe their LFs.

- **Banking77** [5, 23] comprises online banking queries annotated with one of 77 user intents. It has 209 keyword-based LFs.

- **ChemProt** [22] is a chemical relation classification dataset comprising 1,820 PubMed abstracts with chemical-protein interactions annotated by domain experts. The dataset was studied in [2, 49]. Previous works on WS created LFs for the dataset and showed the efficacy of WS in the dataset [45]. We push the boundaries of WS in the dataset by modifying the LFs to incorporate the distance between the chemical and protein entities in the text, among other minor modifications (see Appendix C and Appendix K).

- **Claude9**[4] is based on UNFAIR-ToS [7, 26], which includes 50 Terms of Service (ToS) from online platforms and sentence-level annotations with 8 types of unfair contractual terms (potentially violating user rights according to EU consumer law).[5] LFs for this dataset were created by one of the authors of this work who is a law graduate student.

- **MASSIVE{18,60}** [10, 23] is a corpus of human-to-voice assistant interactions, where the task is to predict one of 60 fine-grained or 18 coarse-grained annotations of user intents. The corpus includes parallel versions across 52 different languages, where each example is present in each language. In our work, we explore reusing the existing LFs written for the English variant of the 18-class prediction task to generalize to additional languages and more fine-grained classes. For non-English variants of MASSIVE18, we apply English LFs by first translating each example back to English using DeepL (see Figure 2). For MASSIVE60, we leverage the original LFs from MASSIVE18 to predict the superclass for each instance. To generalize from coarse to fine-grained predictions in MASSIVE60, we randomly select a subclass corresponding to the predicted superclass from the original LFs.

- **Amazon31** is built from the Amazon product reviews [1] dataset consisting of reviews and their categories.[6] Due to each class's high overlap and conflict rate, we merged several labels and reduced the class cardinality to 31.

**LF Design Pipeline.** We randomly selected samples with clean labels from the training set to create a development set for LFs. We use 250 examples as a development set for Amazon31, which is consistent with the development of LFs for Banking77 and MASSIVE18 in [23]. For Claude9, the development set had 24 examples. We manually inspected labeled examples in the development set, and identified patterns for each class. Then we create multiple keyword-based, dictionary-based, and

---

[3]With the current exception of Amazon31, which was recently retracted, all of the datasets we use are publicly available and with proper licenses (see Appendix B).

[4]Claude9 is imbalanced, with 90%+ of data belonging to one class, so we evaluate using macro-averaged F1.

[5]Art.3 of Direct. 93/13, Unfair Terms in Consumer Contracts (http://data.europa.eu/eli/dir/1993/13/oj).

[6]This dataset has been taken down. We include it in experiments, but we will not release it in BoxWRENCH.

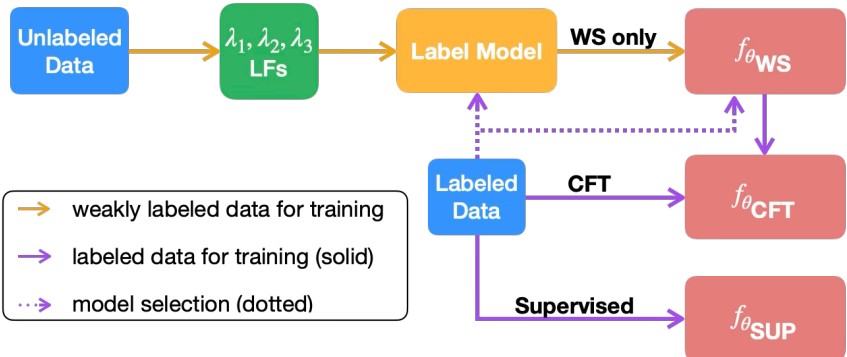

Figure 1: We compare end models trained with three different pipelines: (1) *Supervised*, which uses clean labels from the validation set for fine-tuning; (2) *Weakly Supervised*, which trains on labels obtained by applying all LFs and aggregating them with the label model, using the validation set for hyperparameter search and early stopping. *Continuous-Fine-Tuning*: which takes a weakly supervised model and then continuously fine-tuning it on the same clean validation labels.

regular expression-based LFs for each class[7]. We calculated LF statistics (which do not require label information) on the original training and validation sets, including coverage and LF conflict ratios. To evaluate the final LFs, we calculate their accuracy scores on the original validation set.

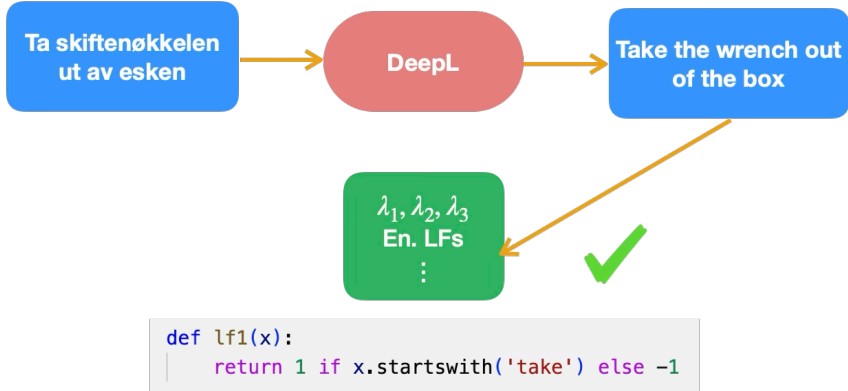

```
def lf1(x):
    return 1 if x.startswith('take') else -1
```

Figure 2: Using an off-the-shelf translator (DeepL) to reuse English LFs for the multilingual variants of MASSIVE18. Our approach is to apply the existing LFs written for MASSIVE18-En by first translating non-English versions of the dataset to English.

## 3.4 Experimental setup

We describe various aspects of our experimental setup: our pipeline, learning scenarios, and a description of how we evaluate when WS is useful.

**Pipeline.** For backward compatibility, our experimental pipeline is based on WRENCH [46]. This allows users to experiment with different LMs, EMs, and LFs in a standardized format, and with WS-specific tooling for dataset manipulation. We use majority vote (MV) as the default LM for most of our WS experiments, following WRENCH's takeaway that MV is one of the most consistent methods across various tasks. Appendix I includes results for COSINE [45] and ARS2 [41], but we did not find that they consistently improved upon MV. That being said, more effective WS methods in the future, would further strengthen our conclusions about WS's effectiveness.

For the end-model, we fine-tune a pretrained RoBERTa model in most cases [27] and also try BERT variants for domain-specific (i.e., LegalBERT[6] for Claude9 and Sci-BERT[4] for ChemProt) and non-English tasks (BERT-base-chinese, NB-BERT-base, and BERT-base-japanese [14, 30, 31]

---

[7]We created the LFs ourselves and did not hire annotators.

for MASSIVE18). Following the procedure used by Zhu et al. [49], we randomly select a set of hyperparameters from their provided hyperparameter search spaces. Whenever training on weak labels, we do early stopping based on the validation data, following the practices of WRENCH, while we fine-tune for a fixed 6,000 total steps whenever training on clean labels, following Zhu et al. [49]. We run all the experiments on NVIDIA A100, A40, A6000, A4000, and RTX-4090 GPUs.

**Learning Scenarios.** We include the three types of learning scenarios that we compare: supervised learning, WS learning, and CFT, all involving clean labels coming from only the validation set (as shown in Figure 1). Notably, we can successfully replicate the results of [49] with our setup on existing WS benchmarks, as shown in Appendix E.

- **Supervised:** We use clean labels from the validation set directly for fine-tuning an end model.
- **Weakly Supervised:** We aggregate all weak labels from the training data with an LM, then fine-tune an end model using the training data with the aggregated labels. Generally in WS, such as in WRENCH, the clean validation labels are used to guide hyperparameter search (for both the LM and end model). In our setup, we use it for early stopping.
- **Continuous-Fine-Tuning (CFT):** We further fine-tune the models trained using only weak super-vision on the same set of clean validation labels.

**Crossover Points.** To measure the usefulness of WS, we plot the performances of the three afore-mentioned methods for various amounts of clean validation data and inspect where their performance curves intersect. Specifically, the crossover point that we care about is between the supervised method, which uses only clean labels, and the better of WS and SFT, which make use of the weak labels.

## 4 Results and Analysis

In this section, we present our main results and analysis. In Section 4.1, we first investigate the crossover points for our new datasets and compare them with existing ones. In Section 4.2, we show that crossover points can be significantly increased when LFs are written more carefully. Section 4.3 showcases another dimension of the usefulness of WS, as we demonstrate how LFs can be adapted in a multilingual setting. Finally, we study whether different LMs perform better on our more challenging new tasks in 4.4.

### 4.1 Comparing crossover points between existing and new benchmarks

We first conducted a crossover point analysis for the existing datasets from WRENCH [46] in Figure 3, extending the results from [49]. We confirm that for most of these tasks, the crossover points between Supervised (green) and CFT (blue) are quite small, less than 200 for four of six tasks. Notably, these four datasets all have considerably smaller label cardinalities compared to most datasets in BOXWRENCH (see Appendix D). We also performed the analysis on the named entity recognition datasets from WRENCH, with similar results (see Appendix H). Overall, these experiments verify that existing benchmarks are often inadequate for capturing realistic scenarios where WS holds significant utility over hand-labeling.

We then conducted the same experiments on our new datasets, and analyzed their crossover points in Figure 4. For both Amazon31 and Banking77, the crossover points are beyond 1,000 clean labels. For Claude9, the validation set is smaller and even training on all available examples does not result in a crossover. Instead, the gap between the CFT and supervised-only methods remains 5% higher.

### 4.2 Improving LFs leads to higher crossovers

Previously, the LFs of ChemProt from Yu et al. [45], which are keywords-based, have a coverage of 0.86 on the test set and a precision of 0.55 (i.e., the accuracy of the set of examples where LM does not abstain). To improve the quality of the LF set, we made the following changes. First, we changed the keywords used in existing LFs. Next, we modified seven LFs by using the absolute difference in entity positions in the text as features. After making these changes, the resulting LF set had a coverage of 0.81 and a precision of 0.63. When constructing these LFs, we consulted a Ph.D student in Neuroscience to leverage his domain expertise in Chemistry and Biology for insights on writing LFs. More details about the new LFs are included in Appendix C.

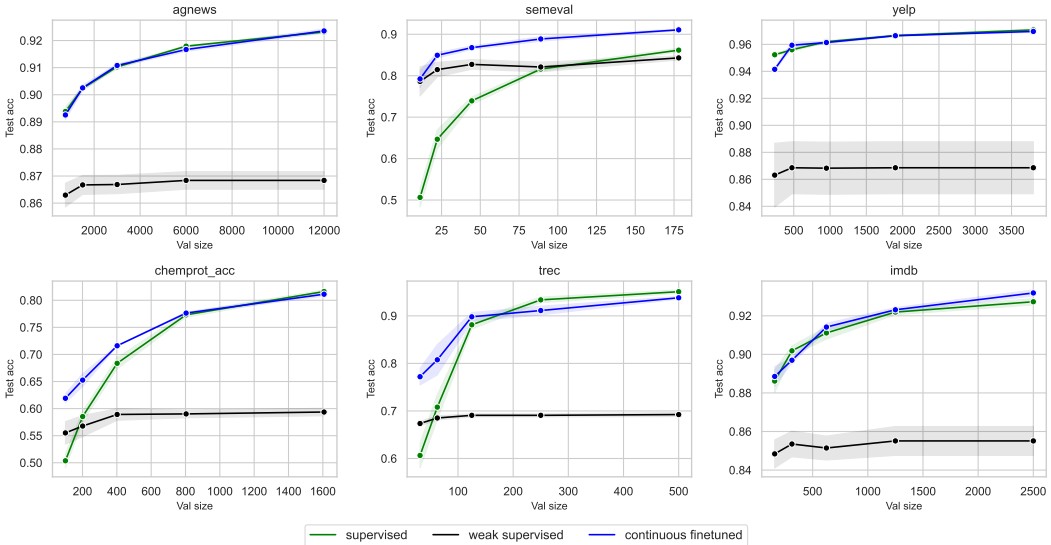

Figure 3: Crossover points on six existing WS benchmarks: AGNews, Semeval, Yelp, ChemProt, Trec, IMDB. The crossover points for these tasks are low, less than 200 on four out of the six tasks, which is substantially lower than the crossover points in BOXWRENCH datasets.

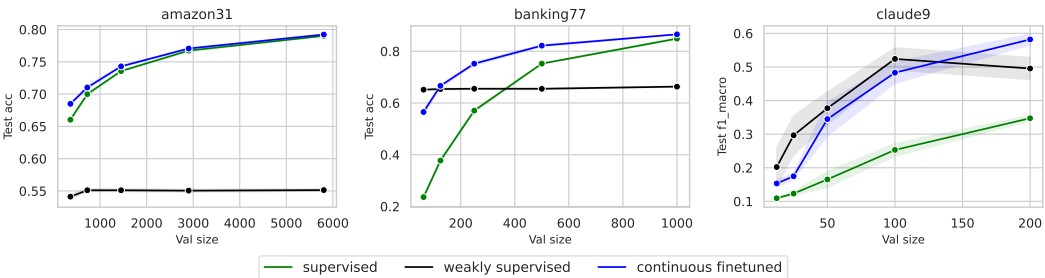

Figure 4: Crossover points on three of our datasets: Amazon31, Banking77, and Claude9. For both Amazon31 and Banking77, the crossover points are beyond 1,000 clean labels. For Claude9, the validation set is smaller and even training on all available examples does not result in a crossover.

These modifications resulted in improved performance for CFT given the same amount of labeled data. When measuring F1, the crossover points grew to around 1600 as shown in Figure 5 (compared to 800 for the original LFs). When measuring accuracy, the improvement was smaller but consistent across the range of validation set sizes (see Figure 7). This experiment demonstrates that more careful writing of LFs can yield higher crossover points. Current research benchmarks may suffer from suboptimal LF sets, and this example highlights how simple adjustments can enhance results.

## 4.3 Cross-lingual adaptability of LFs

In this subsection, we present a set of experiments to highlight an additional advantage of WS over manual labeling: its adaptability. Specifically, we examine the ability to reuse existing LFs in a multilingual setting on the MASSIVE18 dataset. We consider the setting in which one has already trained a weakly supervised model based on LFs written for the dataset's English version and wishes also to obtain a strong model for other languages. In doing so, we consider the following approaches:

- **Oracle-LFs-L** (for L ∈ {Chinese, Japanese, Norwegian}). Here, we directly use the label matrix induced by the LFs for the English version of MASSIVE18 on the non-English versions of the dataset. Note this is only possible because MASSIVE maintains one-to-one correspondences between the examples from different languages.

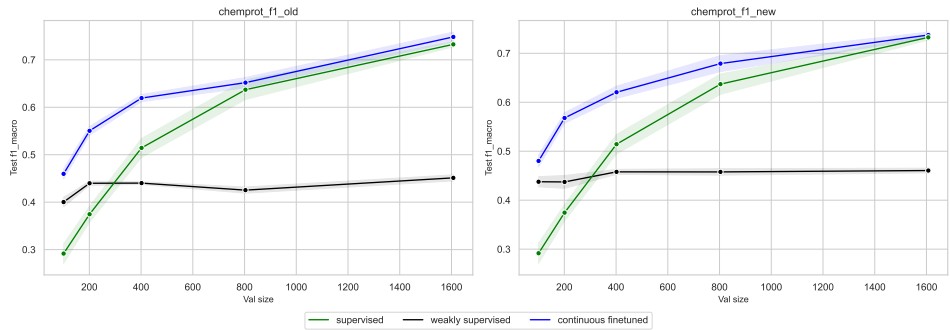

Figure 5: Crossover points on ChemProt with our new LFs. The new LFs for ChemProt demonstrate a higher crossover point when measuring F1 performance.

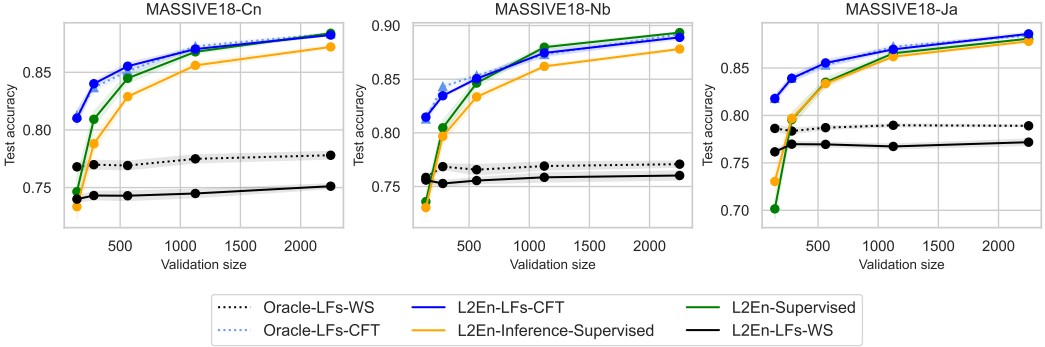

Figure 6: Crossover points on the multilingual MASSIVE18 dataset. The comparison of the green and yellow solid lines showed the importance of having a language-specific model. Surprisingly, using our method from Figure 2 (blue solid line) is able to achieve the same or even better performance compared to the light blue dotted line, which used the weak labels directly from the English version of MASSIVE18.

- **L2En-LFs**, this method intends to provide a more realistic scenario for reusing the English LFs. In practical scenarios, we might have a set of English LFs $LF_{EN}$ and need to train a model in another language, L, without the resources to create additional LFs in a foreign language. In this case, we use DeepL [9] to first translate our non-English unlabeled data to English and then apply $LF_{EN}$ to obtain weak labels. Figure 2 illustrates this pipeline.

- **L2En-Inference-Supervised.** This is an important baseline where we evaluate whether the English model alone can be directly used in other languages by simply first translating test examples into English. In Figure 6, the validation size for this method represents the amount of clean data used to train the English model.

Here we present the accuracy of $f_{\theta_{WS}}$, $f_{\theta_{CFT}}$, and $f_{\theta_{SUP}}$ with our methods in Figure 6, where different suffixes are used with each method for clarity. We use L2En-Inference-Supervised as an additional baseline, representing the use of non-language-specific models, which we observe underperforms training language-specific models (while also incurring additional costs at inference time).

Meanwhile, we see that L2En-LFs demonstrates promise of adapting the English LFs, leading to crossover points of over 1000 in Chinese and Japanese, despite coming for "free" (i.e., without writing additional LFs in the target languages). It also achieves accuracy comparable to Oracle-LFs.

Finally, we also introduce a realistic setup in which even the amount of *unlabeled* data is limited for a rarer language, Norwegian. Specifically, we assume only a 20% subset of the Norwegian version of the MASSIVE is available, which we label using L2En-LFs. We then *augment* this data by translating the corresponding weakly labeled English examples into Norwegian for the remaining 80%. As shown in Figure 9, this leads to a crossover points of above **1500** for fine-tuning Norwegian BERT

Table 2: Test accuracy (ChemPort, MASSIVE18)/F1 scores (Claude9) of supervised-only methods and CFT across different proportions of clean data used, LMs, and datasets. **First** and second best results are **Bolded** and underlined.

|  | Claude9 | ChemProt[8] | MASSIVE18 |
|---|---|---|---|
| **6.25% Validation Size** | 12 | 100 | 127 |
| +Majority vote | 0.153±0.026 | **0.625±0.018** | **0.811±0.004** |
| +DawidSkene | **0.153±0.025** | 0.610±0.020 | 0.797±0.010 |
| +Snorkel | 0.132±0.022 | 0.617±0.012 | 0.811±0.007 |
| +FlyingSquid | 0.109±0.005 | 0.606±0.021 | 0.751±0.011 |
| +Supervised Only | 0.109±0.005 | 0.504±0.030 | 0.753±0.014 |
| **12.5% Validation Size** | 25 | 200 | 254 |
| +Majority vote | 0.175±0.020 | **0.685±0.010** | 0.845±0.006 |
| +DawidSkene | **0.184±0.022** | 0.654±0.024 | 0.836±0.004 |
| +Snorkel | 0.157±0.016 | 0.649±0.015 | **0.845±0.005** |
| +FlyingSquid | 0.124±0.014 | 0.640±0.015 | 0.818±0.009 |
| +Supervised Only | 0.123±0.015 | 0.585±0.022 | 0.820±0.013 |
| **25% Validation Size** | 50 | 401 | 508 |
| +Majority vote | 0.345±0.114 | **0.726±0.021** | 0.863±0.004 |
| +DawidSkene | **0.362±0.107** | 0.712±0.014 | **0.863±0.007** |
| +Snorkel | 0.303±0.090 | 0.711±0.016 | 0.859±0.002 |
| +FlyingSquid | 0.170±0.025 | 0.696±0.010 | 0.856±0.009 |
| +Supervised Only | 0.165±0.052 | 0.684±0.023 | 0.855±0.010 |
| **50% Validation Size** | 100 | 803 | 1016 |
| +Majority vote | **0.483±0.071** | **0.778±0.005** | 0.882±0.005 |
| +DawidSkene | 0.477±0.074 | 0.775±0.003 | 0.884±0.005 |
| +Snorkel | 0.462±0.065 | 0.767±0.009 | **0.885±0.005** |
| +FlyingSquid | 0.232±0.057 | 0.762±0.008 | 0.880±0.005 |
| +Supervised Only | 0.253±0.043 | 0.773±0.010 | 0.883±0.004 |
| **100% Validation Size** | 200 | 1607 | 2033 |
| +Majority vote | **0.582±0.036** | **0.820±0.004** | **0.899±0.002** |
| +DawidSkene | 0.572±0.038 | 0.820±0.005 | 0.893±0.002 |
| +Snorkel | 0.557±0.020 | 0.814±0.004 | 0.899±0.002 |
| +FlyingSquid | 0.352±0.037 | 0.813±0.006 | 0.894±0.005 |
| +Supervised Only | 0.347±0.020 | 0.816±0.007 | 0.898±0.003 |

(NB-BERT-base) [30]. This showcases that for rarer languages, for which foundation models may be less powerful and labels harder to obtain, adapting LFs can be of even larger value.

## 4.4 Label model ablations

In real-world applications, WS is typically integrated with different LMs to optimize performance. Several LMs are frequently employed in WS frameworks, including Majority Vote, Dawid-Skene [8], Snorkel [35], and FlyingSquid [13].

In our study, we conducted a comprehensive evaluation of WS using these LMs on our datasets. The performance of each LM was systematically assessed to determine its effectiveness in various scenarios. Detailed results of this evaluation are presented in Table 2, showcasing the comparative performance and highlighting the strengths of each model (see Appendix F for the full table with Amazon31 and Banking77). For most of the new datasets, the CFT method outperforms the supervised-only method with a clear margin, especially in low-resource settings. Majority Vote and Dawid-Skene performed the best among the label models tested.

---

[8]Here we use ChemProt with updated LFs.

# 5    Limitations

There are several limitations to this work. (1) We primarily focus on text classification tasks, which are more common in practice; while other benchmarks such as WRENCH Zhang et al. [46] include both text classification and sequence-tagging. Similar investigations and LF improvements would be valuable to study using our codebase in future work. (2) In our experiments, we used BERT-based end models and relatively simple label models such as Majority Vote, Dawid-Skene, Snorkel, COSINE, and ARS2. More recent end models and label models could also be worth testing with our pipeline. (3) We did not thoroughly tune hyperparameters while training weakly supervised models, following the setup from [49]. With more careful hyperparameter tuning, WS has the potential to achieve better results. (4) The MASSIVE dataset has one-to-one correspondences across languages, while in real-life scenarios, usage patterns and distribution shifts may exist across different languages, even on identical tasks.

# 6    Conclusions

In this paper, we introduce BOXWRENCH, a benchmark that expands the evaluation of WS by addressing the limitations of existing benchmarks. By incorporating high-class cardinality, imbalance, and the need for domain expertise, BOXWRENCH better reflects real-world data and tasks. Our results show that on these more realistic tasks, weak supervision demonstrates significant utility, particularly in scenarios where traditional labeling is cost-prohibitive. We also show that careful LF design and adapting existing LFs in multilingual settings can significantly enhance the applicability of WS across diverse contexts. BOXWRENCH sets a new standard for evaluating WS, with publicly released datasets, benchmarks, and tools to advance WS research and its practical deployment.

## Acknowledgements.

We are grateful for the support of the NSF under CCF2106707 (Program Synthesis for Weak Supervision) and the Wisconsin Alumni Research Foundation (WARF). Neel Guha is supported by the Stanford Interdisciplinary Graduate Fellowship. We thank Jinoh Lee for his contribution to LF improvements on ChemProt.

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

## A   Broader Social Impact of WS Benchmarking

Our benchmark aims to provide a platform to evaluate WS methods on more realistic datasets. Methods with successful performance are more likely to be useful in real applications, thus improving the effectiveness of WS and reducing the potential costs for practitioners wishing to train ML models. Of course, a potential negative societal impact is that if models are easier to train with WS, people with malicious intent can also train models at a relatively lower cost, leading to potentially harmful impacts.

## B   Dataset Licenses

Our license is CC BY 4.0 license and otherwise inherits the licensing of original datasets.

- **Banking77:** CC-BY-4.0
  https://huggingface.co/datasets/legacy-datasets/banking77
- **ChemProt:** Apache-2.0
  https://github.com/JieyuZ2/wrench
- **Claude9:** CC-BY-4.0
  https://huggingface.co/datasets/coastalcph/lex_glue
- **MASSIVE18:** CC-BY-4.0
  https://huggingface.co/datasets/AmazonScience/massive
- **Amazon31:** No longer available publicly
  https://huggingface.co/datasets/defunct-datasets/amazon_us_reviews

## C   LF Improvements for Chemprot

We described the details of how we improved the LFs for the ChemProt dataset in this document. Further details can be found in our codebase.

### C.1   Original LFs

The original LFs have a coverage of $0.864$ and precision of $0.551$ on the covered data, with an accuracy of 0.4904 using Majority Vote and random tie-breaking (WRENCH reported their accuracy in this way). These statistics are obtained from the test set.

We sampled a development set of size 250 from the training set, examined the definition of each label, and carefully reviewed examples of each label to understand the characteristics of the dataset.

Example LFs are shown below, with the full set found at lable_function/chemprot.

```
# chemprot functions examples:

#0
@labeling_function()
def lf_amino_acid(x):
    return 0 if 'amino acid' in x.text.lower() else ABSTAIN

...

#19
## Cofactor
@labeling_function()
def lf_cofactor(x):
    return 7 if 'cofactor' in x.text.lower() else ABSTAIN

...
```

### C.2   LF Improvement Details

We first started by adding space around or before the keywords in some LFs: "activat", "increas", "reduc", "antagon", "transport", "catalyz", "produc", and "not". This is because for keywords such as "not", they might be triggered by words like "notable".

We also removed LFs with low accuracy on the development set. For example, we removed the function `lf_induce`, as the word "induce" is too general.

Additionally, we developed a utility function, `chemprot_enhanced`, to extend the ChemProt dataframe in WRENCH format with two more columns: `entity1_index` and `entity2_index`. We improved our LFs to utilize these indices to check whether certain words occur between or near the two entities.

After these improvements, on the development set, our coverage dropped to 0.828, but the accuracy for covered data increased to 0.5942. Accuracy with Majority Vote and random tie-breaking rose to 0.508.

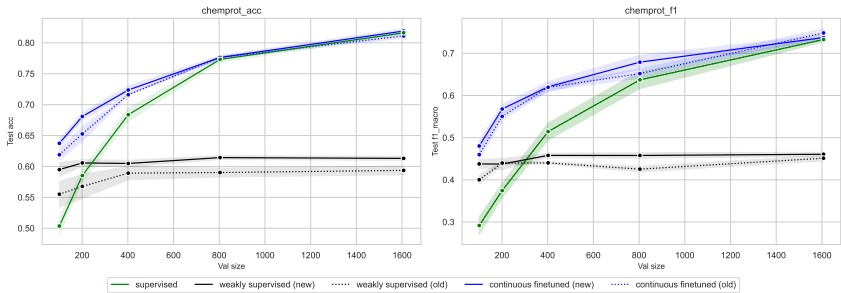

Figure 7: This plot illustrates the crossover points when comparing ChemProt performance with our old and new LFs. The new LFs for ChemProt demonstrate a higher crossover point when measuring F1 performance and smaller (but consistent) gains in accuracy.

# D   Cardinality for WS tasks

Table 3: Number of classes for datasets in WRENCH (top) compared to the new tasks in BOXWRENCH(bottom)

| Dataset | Number of Classes |
|---|---|
| IMDB | 2 |
| ChemProt | 10 |
| TREC | 6 |
| Yelp | 2 |
| SemEval | 9 |
| AGNews | 4 |
| Banking77 | 77 |
| Claude9 | 9 |
| MASSIVE18 | 18 |
| MASSIVE60 | 60 |
| Amazon31 | 31 |

# E Reproducing results from Zhu et al. [49].

Here, we show how using our codebase, we can successfully reproduce the results from Zhu et al. [49].

| Dataset | N | Implementation | Supervised | Weakly Supervised | CFT |
|---|---|---|---|---|---|
| AGNEWS | 50 | Zhu et al. [49] | 0.880 | 0.872 | 0.882 |
| | 5 | Zhu et al. [49] | 0.770 | 0.840 | 0.841 |
| | 50 | Ours | $0.875 \pm 0.007$ | $0.859 \pm 0.010$ | $0.871 \pm 0.017$ |
| | 5 | Ours | $0.769 \pm 0.040$ | $0.863 \pm 0.009$ | $0.820 \pm 0.030$ |
| Yelp | 50 | Zhu et al. [49] | 0.950 | 0.820 | 0.910 |
| | 5 | Zhu et al. [49] | 0.740 | 0.760 | 0.840 |
| | 50 | Ours | $0.947 \pm 0.006$ | $0.868 \pm 0.043$ | $0.943 \pm 0.075$ |
| | 5 | Ours | $0.767 \pm 0.056$ | $0.843 \pm 0.072$ | $0.915 \pm 0.014$ |
| IMDb | 50 | Zhu et al. [49] | 0.880 | 0.818 | 0.864 |
| | 5 | Zhu et al. [49] | 0.705 | 0.795 | 0.797 |
| | 50 | Ours | $0.868 \pm 0.030$ | $0.846 \pm 0.023$ | $0.889 \pm 0.007$ |
| | 5 | Ours | $0.630 \pm 0.064$ | $0.819 \pm 0.027$ | $0.794 \pm 0.054$ |
| TREC | 50 | Zhu et al. [49] | 0.930 | 0.680 | 0.940 |
| | 5 | Zhu et al. [49] | 0.630 | 0.640 | 0.840 |
| | 50 | Ours | $0.911 \pm 0.014$ | $0.678 \pm 0.097$ | $0.910 \pm 0.013$ |
| | 5 | Ours | $0.603 \pm 0.046$ | $0.662 \pm 0.036$ | $0.815 \pm 0.040$ |
| ChemProt | 50 | Zhu et al. [49] | 0.720 | 0.550 | 0.730 |
| | 5 | Zhu et al. [49] | 0.420 | 0.510 | 0.590 |
| | 50 | Ours | $0.707 \pm 0.0163$ | $0.583 \pm 0.012$ | $0.737 \pm 0.0069$ |
| | 5 | Ours | $0.420 \pm 0.023$ | $0.518 \pm 0.030$ | $0.573 \pm 0.027$ |
| SemEval | 50 | Zhu et al. [49] | 0.862 | 0.820 | 0.910 |
| | 5 | Zhu et al. [49] | 0.720 | 0.760 | 0.840 |
| | 50 | Ours | $0.855 \pm 0.0037$ | $0.837 \pm 0.016$ | $0.916 \pm 0.077$ |
| | 5 | Ours | $0.747 \pm 0.021$ | $0.836 \pm 0.006$ | $0.868 \pm 0.006$ |

# F Extended Table for Section 4.4

Table 4 extends the results in Section 4.4, showing additional results for Amazon31 and Banking77.

Table 4: Additional test accuracy for Amazon31 and Banking77. Best results are **Bolded**.

| | 6.25% Validation | 12.5% Validation | 25% Validation | 50% Validation | 100% Validation |
|---|---|---|---|---|---|
| **Banking77** | | | | | |
| +Majority vote | **0.565±0.026** | **0.667±0.015** | **0.752±0.015** | **0.822±0.010** | **0.865±0.002** |
| +Supervised Only | 0.236±0.016 | 0.378±0.020 | 0.571±0.018 | 0.752±0.004 | 0.849±0.007 |
| **Amazon31** | | | | | |
| +Majority vote | **0.685±0.005** | **0.710±0.007** | **0.743±0.003** | **0.771±0.003** | **0.792±0.002** |
| +Supervised Only | 0.660±0.008 | 0.700±0.006 | 0.736±0.003 | 0.767±0.003 | 0.790±0.001 |

# G MASSIVE60 and Using English for Augmentation

This section contains the results for MASSIVE60 (Figure 8), as well as an exploratory approach for leveraging non-English data as additional weak supervision for improving performance on MASSIVE-En. Section 4.3 (Figure 9).

MASSIVE60, with our straightforward methodology, demonstrates a crossover point exceeding 500, highlighting its potential for the application of WS techniques.

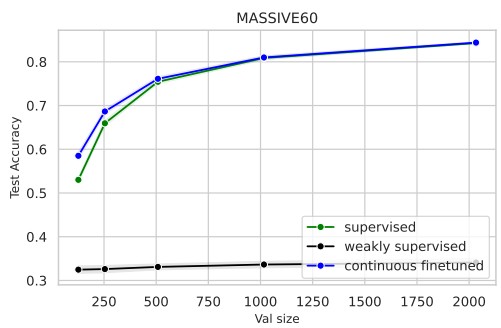

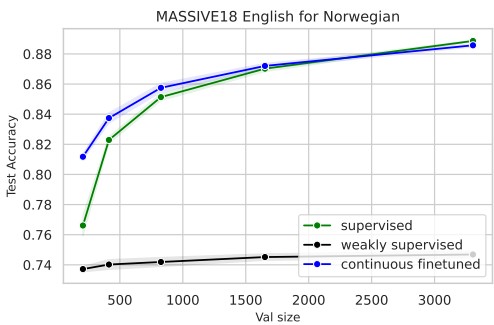

Figure 8: MASSIVE60

Figure 9: Using English text as an augmented data for Norwegian BERT

## H   NER datasets

We included additional experiments related to named entity recognition datasets (BC5CDR[25], NCBI-Disease[11], CoNLL-03[38], OntoNotes 5.0[44]) at Figure 11. Overall, we observe that crossover points are relatively small but note that each sequence has multiple label entities, which would further increase the cost of manual labeling compared to sequence classification tasks. Finding tasks and LFs for NER that have higher crossover points would be an interesting avenue for future work.

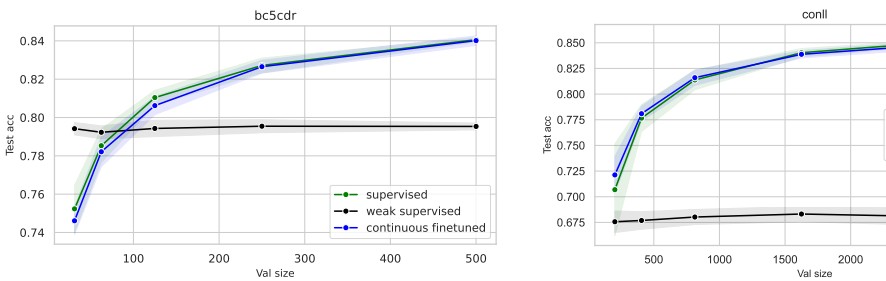

Figure 10: Crossoverpoints for additional NER task.

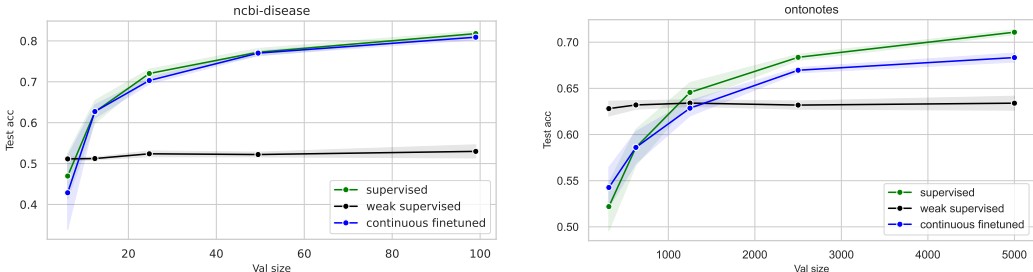

Figure 11: Crossoverpoints for additional NER task.

## I   Cosine + ARS2

We experimented with an additional weak supervision baseline with the same experimental pipeline. We include results for COSINE and ARS2 in Figure 12. We used both of the learning methods with a RoBERTa/Legal BERT backbone on the WS-only pipeline. The other setups were kept the same. The ARS2 results follow a similar trend to our previous results, with almost identical cross-over

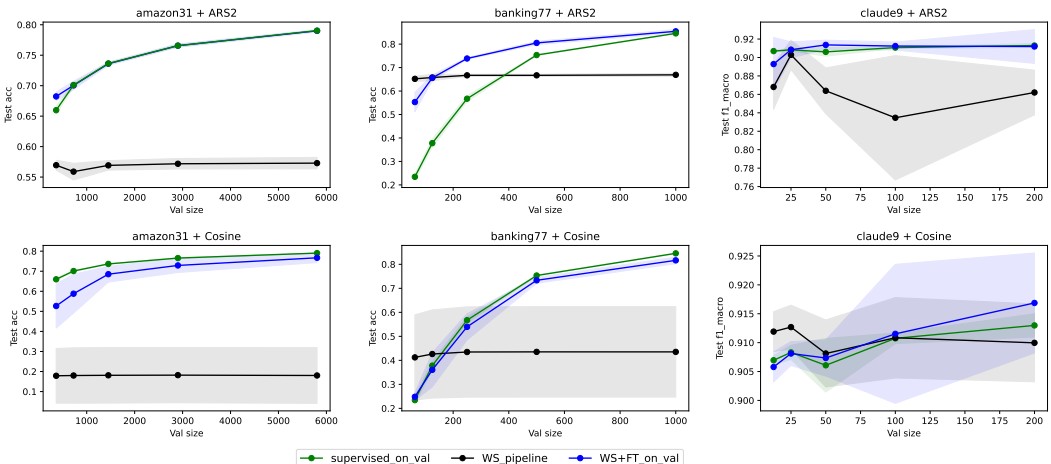

Figure 12: Crossover points using ARS2 and Cosine as end models.

points. COSINE performance is slightly worse on Amazon31 and Banking77, and is slightly better on Claude9, when compared with the supervised-only method.

## J  F1 scores for all the dataset

Since some of the datasets are highly imbalanced, we provide other metrics (micro, macro, weighted F1 scores) for those Table 5, providing a more complete picture of performance across imbalanced classes. The inclusion of these metrics does not change any of our conclusions.

## K  SciBERT

We used in-domain SciBERT[4], a pretrained language model for scientific text to test whether the crossover point is still consistent. The crossover point is even higher in this case, suggesting the usefulness of weak supervision, see Figure 13.

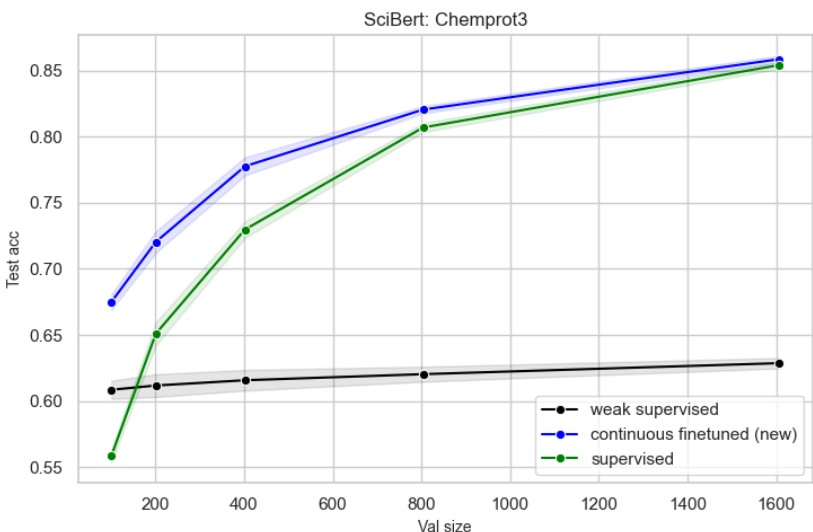

Figure 13: SciBERT Results

Table 5: Accuracy and F1 scores for different datasets and methods with varying validation sizes (VS). Best results are **bolded**.

| | 6.25% VS | 12.5% VS | 25% VS | 50% VS | 100% VS |
|---|---|---|---|---|---|
| **Amazon31** | | | | | |
| +CFT (Accuracy) | **0.685** | **0.711** | **0.742** | **0.769** | **0.791** |
| +Supervised Only (Accuracy) | 0.660 | 0.701 | 0.736 | 0.766 | 0.790 |
| +CFT (F1_micro) | **0.685** | **0.711** | **0.742** | **0.769** | **0.791** |
| +Supervised Only (F1_micro) | 0.660 | 0.701 | 0.736 | 0.766 | 0.790 |
| +CFT (F1_macro) | **0.683** | **0.710** | **0.741** | **0.768** | **0.791** |
| +Supervised Only (F1_macro) | 0.659 | 0.702 | 0.736 | 0.766 | 0.790 |
| +CFT (F1_weighted) | **0.682** | **0.709** | **0.740** | **0.767** | **0.790** |
| +Supervised Only (F1_weighted) | 0.658 | 0.701 | 0.735 | 0.765 | 0.789 |
| **Banking77** | | | | | |
| +CFT (Accuracy) | **0.551** | **0.676** | **0.751** | **0.820** | **0.866** |
| +Supervised Only (Accuracy) | 0.234 | 0.378 | 0.567 | 0.753 | 0.846 |
| +CFT (F1_micro) | **0.551** | **0.676** | **0.751** | **0.820** | **0.866** |
| +Supervised Only (F1_micro) | 0.234 | 0.378 | 0.567 | 0.753 | 0.846 |
| +CFT (F1_macro) | **0.497** | **0.650** | **0.742** | **0.818** | **0.865** |
| +Supervised Only (F1_macro) | 0.167 | 0.318 | 0.530 | 0.745 | 0.844 |
| +CFT (F1_weighted) | **0.497** | **0.650** | **0.742** | **0.818** | **0.865** |
| +Supervised Only (F1_weighted) | 0.167 | 0.318 | 0.530 | 0.745 | 0.844 |
| **Claude9** | | | | | |
| +CFT (Accuracy) | 0.904 | 0.906 | **0.906** | **0.914** | **0.914** |
| +Supervised Only (Accuracy) | **0.907** | **0.908** | 0.906 | 0.911 | 0.913 |
| +CFT (F1_micro) | 0.904 | 0.906 | **0.906** | **0.914** | **0.914** |
| +Supervised Only (F1_micro) | **0.907** | **0.908** | 0.906 | 0.911 | 0.913 |
| +CFT (F1_macro) | **0.149** | **0.177** | **0.351** | **0.493** | **0.558** |
| +Supervised Only (F1_macro) | 0.110 | 0.120 | 0.173 | 0.256 | 0.349 |
| +CFT (F1_weighted) | **0.874** | **0.877** | **0.895** | **0.910** | **0.916** |
| +Supervised Only (F1_weighted) | 0.864 | 0.866 | 0.870 | 0.884 | 0.900 |
| **ChemProt** | | | | | |
| +CFT (Accuracy) | **0.637** | **0.681** | **0.724** | **0.776** | **0.819** |
| +Supervised Only (Accuracy) | 0.595 | 0.606 | 0.605 | 0.614 | 0.613 |
| +CFT (F1_micro) | **0.637** | **0.681** | **0.724** | **0.776** | **0.819** |
| +Supervised Only (F1_micro) | 0.595 | 0.606 | 0.605 | 0.614 | 0.613 |
| +CFT (F1_macro) | **0.453** | **0.523** | **0.572** | **0.640** | **0.710** |
| +Supervised Only (F1_macro) | 0.405 | 0.410 | 0.412 | 0.409 | 0.412 |
| +CFT (F1_weighted) | **0.623** | **0.669** | **0.715** | **0.770** | **0.816** |
| +Supervised Only (F1_weighted) | 0.573 | 0.581 | 0.581 | 0.587 | 0.586 |

# L    Comparison with or using LLMs

LLMs have demonstrated strong zero-shot or few-shot capabilities, one may also curious about how SOTA LLMs perform on our datasets. We sampled 250 data points from our test set, utilized API calls with GPT-4o-2024-08-06 for each example, and recorded the accuracy. We attached the cross-over points graph with this baseline (See Figure 14). The LLM baseline performed poorly in tasks requiring domain-specific knowledge, such as Claude9 and ChemProt. It performed well on Amazon31, possibly because the test set may be included in the public amazon review datasets, which GPT may have been trained on.

We also prompted GPT-4o as if it was a domain expert (e.g. "you are an expert in legal document classification and label function writing") and asked it to create keywords-based LFs based upon a development set. The full list of our prompts is included in our codebase. While the coverage of the GPT-4o-generated label functions is higher, the precision of the LFs is generally lower (see Table 6). However, we note that the generated labeling functions (LFs) are acceptable, especially considering

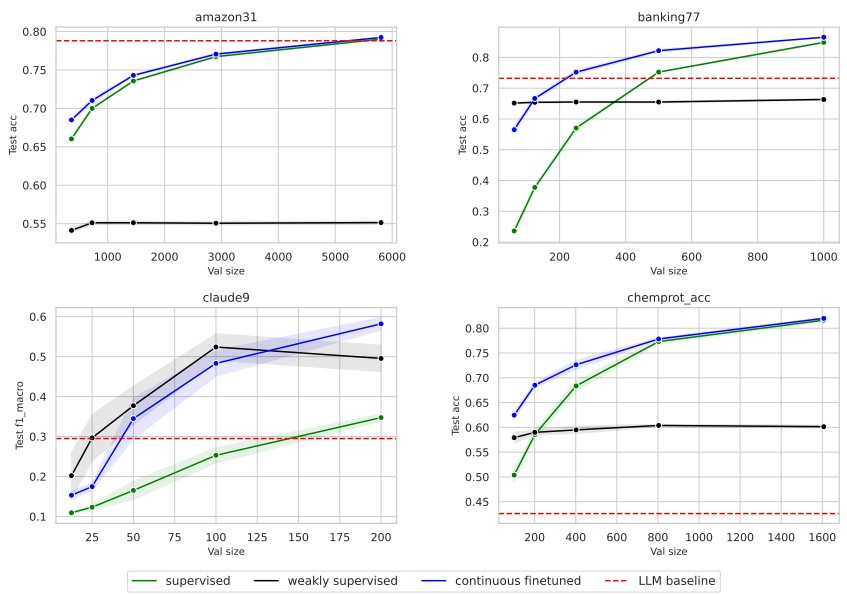

Figure 14: Using LLM as baseline

the reduction in time cost and the potential for full automation. Future work could involve analyzing a multi-agent scenario that automates and reinforces the LF generation process.

Table 6: Coverage and Precision Comparison Between Our LFs and LLM-Generated LFs

| Dataset | Our Coverage | Our Precision | LLM Coverage | LLM Precision |
|---|---|---|---|---|
| Amazon31 | 0.63 | 0.70 | 0.99 | 0.26 |
| Banking77 | 0.58 | 0.81 | 0.98 | 0.62 |
| ChemProt | 0.82 | 0.70 | 0.96 | 0.56 |
| Claude9 | 1.00 | 0.91 | 0.91 | 0.15 |

# M  Miscellaneous

## M.1  Links to datasets & metadata

We noted that all the datasets that we used are based on previously publicly published datasets.
The license and links are mentioned in Appendix B. In addition to the original link for the datasets
mentioned in Appendix B. We also provide our own usage of the datasets on GoogleDrive.

## M.2  Dataset Format

The dataset uses the same format as the WRENCH [46] benchmark. For each dataset directory, there
are four `.json` files for training data, validation data, test data, and labels respectively. For the label
file, the labels are organized in the following format:

```
{
Label index: Label name,
...
}
```

For the dataset files, the data points are organized in the following format:

```
{
Data index: {
    "labels": Label index,
    "weak_labels": [-1, -1, ...],
    "data": {
        "text": Data Content,
        ...,
        More content depending on the datasets
    }
}
}
```

## M.3  Structured Metadata

We also provide our datasets on HuggingFace, and the metadata are contained in the **README.md**
for each dataset.

