# Appendix

The appendix is organized as follows. First, we include the licenses and links to the dataset. Then we provide the cardinality of the old WRENCH dataset for reference. In the following sections, we further discuss the limitation and the broader societal impact on our public respectively. Next, we include the result reported from Zhu et al. [2023] and an extended table for Section 4.4. Next, we described all the existing pipeline functionality of our code-base. We also presented the graph of massive multi-languages. Which showed the crossover points for Chinese, Norwegian, and Japanese.

## A    Dataset Licenses

The license information for each dataset is provided below.

- **Banking77:** CC-BY-4.0
  https://huggingface.co/datasets/legacy-datasets/banking77
- **ChemProt:** Apache-2.0
  https://github.com/JieyuZ2/wrench
- **Claude9:** CC-BY-4.0
  https://huggingface.co/datasets/coastalcph/lex_glue
- **Massive18:** CC-BY-4.0
  https://huggingface.co/datasets/AmazonScience/massive
- **Amazon31:** No longer available publicly
  https://huggingface.co/datasets/defunct-datasets/amazon_us_reviews

## B    Cardinality of WRENCH Dataset

- IMDB: 2
- ChemProt: 10
- TREC: 6
- Yelp: 2
- Semeval: 9
- AgNews: 4

## C    Limitations of Our Work

There are several limitations to this work.

- The authors who wrote the LFs for the dataset are computer science students with little to no background in the corresponding fields such as chemical engineering or finance. Thus, further improvement can be made with professionals with domain expertise.
- This paper only primarily evaluated the performance of Roberta's (some other BERT variant) EM. Similarly, this paper only tested the performance of some relatively simple LMs such as Majority Vote, Dawid-Skene, and Snorkel. More complex and recent models could be tested using the pipeline.
- This paper fixed the number of steps and hyperparameter while training following settings from Zhu et al. [2023]. With proper early stopping and hyperparameter search, WS has the potential to achieve better results.
- The Massive dataset has one-to-one correspondence across languages, while in real-life scenarios, distribution shifts among different languages even on identical tasks. Thus multi-lingual dataset collected from real uses would be more effective in evaluating the performance

Table 3: Additional test accuracy for Amazon31 and Banking77. Best results are **Bolded**.

| | 6.25% VALIDATION | 12.5% VALIDATION | 25% VALIDATION | 50% VALIDATION | 100% VALIDATION |
|---|---|---|---|---|---|
| **BANKING77** | | | | | |
| +MAJORITY VOTE | **0.5652±0.0262** | **0.6665±0.0152** | **0.7519±0.0150** | **0.8218±0.0098** | **0.8654±0.0023** |
| +SUPERVISED ONLY | 0.2362±0.0155 | 0.3777±0.0199 | 0.5705±0.0176 | 0.7522±0.0041 | 0.8488±0.0065 |
| **AMAZON31** | | | | | |
| +MAJORITY VOTE | **0.6850±0.0046** | **0.7103±0.0066** | **0.7429±0.0028** | **0.7706±0.0032** | **0.7923±0.0016** |
| +SUPERVISED ONLY | 0.6603±0.0078 | 0.6999±0.0056 | 0.7357±0.0026 | 0.7673±0.0033 | 0.7902±0.0012 |

# D   Broader Social Impact of WS Benchmarking

This work would have a positive societal impact. This benchmark provides a platform to evaluate the WS method on a more realistic dataset. Methods with successful performance are more likely to be useful in real applications, thus improving the effectiveness of the WS model training process and reducing the cost and resources required.

However, a potential negative societal impact is that since all of our datasets and codes are publicly available, people with malicious intent can easily use the datasets. In addition, WS makes it possible for these individuals to train models at a relatively lower cost, leading to potentially harmful impacts.

# E   Reproducing Zhu et al. [2023].

We reproduce the results in Zhu et al. [2023], we also pointed out that their graphs may be misleading. In the graph, they claim that their data are stratified, while for several datasets only the unstratified data can achieve the same accuracy. The graphs claim that the data are stratified; however, for several datasets, only the unstratified data achieves the reported accuracy. Although this discrepancy is mentioned in a small part of their text, it seems to have been obscured, and the text on the graph does not clarify this.

We present our reproduction of their results in Table 4.

# F   Extended Table for Section 4.4

This is an extension to the table in the Section 4.4 with some results on Amazon31 and Banking77. See Table 3.

# G   Pipeline Functionality

- **oracle**: It uses all the training data of the dataset with ground truth labels and trains a supervised model on it.
- **end-to-end**: It uses all the labeled validation data and unlabeled training data. It first trains a weakly supervised model to create weak labels for all the training data, and aggregates to get strong labels that get piped into a fully supervised end model.
- **val-as-train**: It takes a range of numbers/proportions of the labeled validation data and trains a supervised model just on those subsampled validation data.
- **train-as-train**: It takes a range of numbers/proportions of the training data (with ground truth), and trains a supervised model just on those subsampled validation data.
- **saturation** These experiments require orcale experiment of the same dataset and supervised model to be run first. Then it performs a binary search on the number/proportion of labelled data required to match the oracle performance.
- **fine-tune-on-val** This experiment is based on the end-to-end experiment, and performs an additional step of fine-tuning on the trained supervised model with the labeled validation dataset.

Table 4: Replication Results

| Dataset | Paper N = 50 (baseline/before/after) | Paper N = 5 (baseline/before/after) | Ours N = 50 (baseline/before/after) | Ours N = 5 (baseline/before/after) |
|---|---|---|---|---|
| AGNEWS | 88/87.2/88.2 | 77/84/84.1 | 87.5(0.7)/85.9(1)/87.1(1.7) | 76.9(4)/86.3(0.9)/82(3) |
| Yelp | 95/82/91 | 74/76/84 | 94.7(0.6)/86.8(4.3)/94.3(7.5) | 76.7(5.6)/84.3(7.2)/91.5(1.4) |
| IMDb | 88/81.8/86.4 | 70.5/79.5/79.7 | 86.8(3)/84.6(2.3)/88.9(0.7) | 63(6.4)/81.9(2.7)/79.4(5.4) |
| TREC | 93/68/94 | 63/64/84 | 91.1(1.4)/67.8(9.7)/91.04(1.3) | 60.3(4.6)/66.2(3.6)/81.5(4) |
| ChemProt | 72/55/73 | 42/51/59 | 70.7(1.63)/58.3(1.2)/73.71(0.69) | 42(2.3)/51.8(3)/57.3(2.7) |
| SemEval | 86.2/82/91 | 72/76/84 | 85.5(0.37)/83.7(1.6)/91.6(7.7) | 74.67(2.1)/83.6(0.6)/86.8(0.6) |

## H   Multi-Language Massive

We provide the extended results for the massive datasets in Section 4.3. The results are gathered from 12 datasets, see Figure 7.

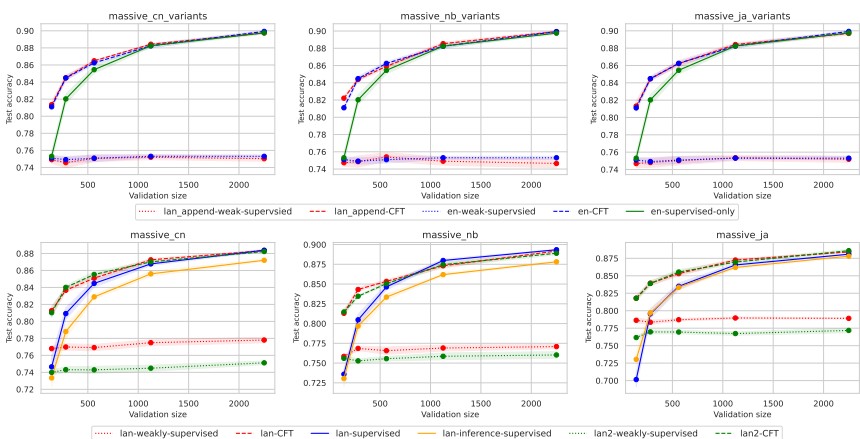

Figure 7: WS for Multi-Language Massive

## I   Massive HC and Using English for Augmentation

This section contains the results for a high cardinality version massive with 60 classes (Figure 8), as well as a more realistic use case involving the English dataset mentioned in Section 4.3 (Figure 9).

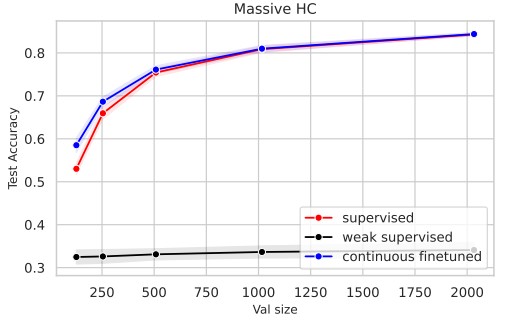

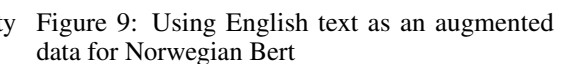

Figure 8: Massive dataset with higher cardinality (60)

Figure 9: Using English text as an augmented data for Norwegian Bert

## J  Links to datasets & metadata

We noted that all the datasets that we used are based on previously publicly published datasets. The license and links are mentioned in Appendix A. In addition to the original link for the datasets mentioned in Appendix A. We also provide our own usage of the datasets on GoogleDrive.

## K  Dataset Format

The dataset uses the same format as the WRENCH Zhang et al. [2021] benchmark. For each dataset directory, there are four `.json` files for training data, validation data, test data, and labels respectively. For the label file, the labels are organized in the following format:

```
{
Label index: Label name,
...
}
```

For the dataset files, the data points are organized in the following format:

```
{
Data index: {
    "labels": Label index,
    "weak_labels": [-1, -1, ...],
    "data": {
        "text": Data Content,
        ...,
        More content depending on the datasets
    }
}
```

## L  Long-term Preservation

The datasets will be hosted indefinitely at the provided link.

## M  Explicit License

Our license is CC BY 4.0 license and otherwise inherits the licensing of original datasets.

## N  Structured Metadata

We also provided our datasets on HuggingFace, and the metadata are contained in the **README.md** for each dataset.

## O  Other Sources

Our code is maintained on GitHub.