# OpenReview forum: "Stronger Than You Think: Benchmarking Weak Supervision on Realistic Tasks"
_NeurIPS.cc/2024/Datasets_and_Benchmarks_Track — NeurIPS 2024 Track Datasets and Benchmarks Poster_

### Official Review · Reviewer_2MXd · 2024-06-29
**Paper 1999 Review**

**Rating:** 5
**Confidence:** 5
**Correctness:** The claim is not well-justified. Plea…
**Clarity:** Not very clear. Critical information …

**Review:**

I think the idea and motivation of this work are potentially interesting to researchers working on similar topics, but the paper does not support its title and conclusion in a very convincing way. My main concerns include the following points.

1. From my understanding, this paper only addresses the text classification task. However, both Zhang et al. (2021) and Zhu et al. (2023) conducted experiments on both text classification and named entity recognition, the latter of which is generally considered a harder task. It would be advisable to also include such tasks in your discussion to verify your statement.

2. LF improvement is one of the significant contributions claimed by the authors, but there is no detailed discussion on this topic in the paper or the appendix. There should be a section dedicated to LFs' construction and their performance on each dataset with ablation studies and discussions on how each of them will affect the final results. In addition, it is also suggested to compare the time spent in developing such LFs against the time of labeling the datasets manually when a similar test performance is achieved.

3. The limitation section is a bit confusing. First of all, I think it should be put into the main article body instead of the appendix. In addition, the first item of the appendix claims that all LFs are developed by CS students, whereas the article treats the "datasets that require domain knowledge" as a big selling point. It is also confusing why the authors choose RoBERTa for all datasets except for multi-lingual experiments, when it needs almost no effort to choose a proper in-domain pre-trained BERT variant for each dataset for fair comparisons.

4. The authors should report the accuracy, micro&weighted&macro F1 of **all** datasets, considering that the authors claim their datasets are "highly imbalanced".

5. The figures and tables could use some more comprehensive captions.

6. typo across the article: weak supervised -> weakly supervised.

7. I really appreciate the authors sharing their code, but I'd appreciate it more if the code is organized in a clearer structure. I was trying to identify the LF construction functions but intimidated by the randomly placed files. I noticed that there is a commit labeled as "Repo cleaning". Probably the authors can put some more effort in it. This is just a random comment and does not affect my score.

**Strengths:**

Please check the review section.

**Additional Feedback:**

N/A

**Documentation:**

The dataset is not sufficiently discussed.

**Ethics:**

No.

**Limitations:**

Check the "Review" section

**Opportunities For Improvement:**

Check the "Review" section

**Relation To Prior Work:**

Although I think there are plenty of works in weak supervision that are neglected by the article, I believe the authors have included critical ones and discussed the differences.

**Summary And Contributions:**

This paper introduces BoxWRENCH, which extends the previous weak supervision benchmarks such as WRENCH to scenarios with larger numbers of label classes, higher label imbalance, and requirements for domain knowledge. By comparing simple WS methods with fully supervised models, the authors claim that although WS methods fall short in simple cases, as observed by Zhu et al. (2023), they are still competitive when equipped with carefully curated labeling functions. In addition, it is easy to extend the labeling functions to multi-lingual cases.

---

> ### Author Rebuttal · Authors · 2024-08-17
>
> Thank you for the detailed review! We agree with many of your points!
>
> **On named entity recognition.** We focus on the text classification tasks for this benchmark, as it comprises the majority of the tasks used by Zhang et al. (2021),  Roberts et al. (2022), and  Zhu et al. (2023). Nonetheless, sequence tagging is indeed included by Zhang et al. (2021).  Since submission, **we have conducted further experiments on two named entity recognition** tasks from WRENCH as a validation of our pipelines. Specifically, we evaluate OntoNotes 5.0 (Weischedel et al., 2013) and MIT-Movies (Liu et al., 2013). The results for MIT-Movies are presented in Figure 2 of the attached PDF. The cover-over points still occur around 50 labeled data points. The results for OntoNotes will be available in the next few days, as it is a significantly larger dataset, and will be posted during the discussion session.
>
> **On LF details and ablations.** Due to limited space, we did not fully describe all the details. We have also shared the exact notebook we used to develop the LFs in the repo. We agree that it could be useful to study how individual LF performance affects the final result. However, since the LFs for some datasets are keyword-based, conducting a "leave one out" experiment for datasets like Banking 77, which has 209 different LFs, would be computationally expensive. We instead include the procedure for LF improvement, and instructions on further analysis related to coverage, precision, conflict ratio, and accuracy of each individual LF in the common response and our [codebase](https://github.com/jeffreywpli/stronger-than-you-think/tree/main/end_model_training/lable_function).
>
> **On time spent on LF development.** We agree that a comparison between the time spent developing LFs and manual labeling is an important consideration for weak supervision, in general. The time required is approximately a few hours. We will include this detailed information in our new results, which feature newly designed LFs created by a neuroscience student specifically for the ChemProt dataset.
>
> **On the limitations section.** Due to limited space, we included our limitation section in the appendix. We will bring it into the main text in the final version.
>
> **On domain expertise.** We did not make it clear that some of the CS students also have relevant backgrounds to effectively create the LFs for the datasets that “require domain knowledge.” For example, one of our LF sets was developed by a Ph.D. student who, in addition to having CS experience, is also in a law program. Furthermore, we have collaborated with a neuroscience Ph.D. student to produce an alternative LF set for ChemProt for comparison with those created by CS students. The results will be obtained in the next couple of days and we will also include the detailed comparison with our previous LFs. Lastly, with more domain expertise put into the development of LFs for those hard tasks, the result of WS could go beyond just marginally better compared to traditional supervised methods.
>
> **On in-domain BERT variant.** We agree that the in-domain pre-trained BERT variant could improve the results for some of the datasets. We used Legal BERT for the Claude9 dataset and Sci-Bert for ChemProt. Using Legal BERT does not affect the general behavior between CFT and the supervised-only method (Figure 1 in the attached PDF). Results of using Sci-Bert with neuroscience students’ LFs will be posted during the discussion period.
>
> **On imbalanced dataset.** We agree that for imbalanced datasets, F1 scores could provide meaningful insights. We have included additional results for accuracy, micro/weighted/macro F1 for all datasets (Table 1 in the attached PDF). The relative performance ranking between the CFT and the supervised-only method remains consistent across different metrics. To further validate our claim that some of the datasets are “highly imbalanced”, we included the label distribution of our dataset in Figure 3-6 in the attached PDF. It is evident that all datasets, except for Amazon31, exhibit imbalanced class distributions. Although Amazon31 is balanced, it is still a dataset with high cardinality.
>
> **On caption.** Thank you for pointing this out, we will make the captions more descriptive in the final version.
>
> **On inconsistency between “weak supervised” and “weakly supervised”.** Thank you for catching this. We will ensure consistency in the final version.
>
> **On codebase.** Thank you for looking into our codebase. We will release a more user-friendly repo of the codebase in the final version.

---

> > ### Comment · Reviewer_2MXd · 2024-08-18
> > **Response**
> >
> > Thanks for your reply, I have a few further questions and concerns.
> >
> > - **NER** Thanks for attending to this comment. But I think to make the NER cases support your claim, you might also need to put some effort into LF design. The current results from MIT-Movies do not seem promising. Otherwise, you need to specifically mention in some noticeable place that your claim is only valid for weakly supervised sequence classification to avoid confusing others. BTW, why did you select these two datasets rather than the more widely used ones such as CoNLL 2003, or the domain-specific BC5CDR or NCBI-Disease?
> >
> > - **LF Details** This is not a valid excuse. The improved LF design is the most important claim in your paper. You should probably spend around two pages discussing how these LFs are designed, how much time it takes, how much domain expertise they require, how these LFs affect the final performance, etc. You may move other parts into the appendix, but you must have such details in the body.
> >
> > - **Domain Expertise** Your answer contradicts your limitation section (Line 461).
> >
> > I'm overall satisfied with other responses except for the promises, which I can only judge when they come true. But I'll keep my score as I think the important issues are not well-addressed.

---

> > > ### Author Rebuttal · Authors · 2024-08-25
> > >
> > > Thank you for your reply and further questions.
> > >
> > > **On NER.** We thank you for your input. Our work is centered around **text classification**, which is more common in practice. We will update our wording in the final version to clarify the focus better. We also started with MIT-Movies and OntoNotes5.0 because both of these datasets have higher class cardinalities than those suggested: MIT-Movies has 12 classes and OntoNotes5.0 has 18 classes. However, we also note that each sequence has multiple label entities, which would further increase the cost of manual labeling (See Table 1). That being said, thank you for the suggestion to try the other datasets, which we have now finished (See attached PDF). All datasets exhibit similar behavior. The results showcase some potential for LF improvements, which we believe, however, belong to the scope of future work.
> > >
> > > ## Table 1: Dataset and Averaged number of Entity per Sequence
> > > | Dataset         | entity/sequence |
> > > |-----------------|--------|
> > > | OntoNotes5.0       | 16.8974 |
> > > | MIT-Movies       | 10.304 |
> > > | CoNLL           | 15.8    |
> > > | NCBI-Disease    | 226.8   |
> > > | BC5CDR          | 213.4   |
> > >
> > > **On LF Details.** We have carefully considered your suggestions and will incorporate a dedicated section in our paper that provides descriptive details and a thorough analysis of the newly enhanced LFs. We’ve also added a detailed LF improvements explanation at [GitHub](https://github.com/jeffreywpli/stronger-than-you-think/tree/main/end_model_training/lable_function/chemprot). **We also would like to clarify that while improved LF design is one aspect of our work, it is not the sole focus.** Our primary goal is to create **better and more realistic benchmarks**, rather than improving or crafting LF sets with SOTA performance. Additionally, our paper also highlights other important aspects, such as the generalizability of WS across different languages and the usefulness of WS.
> > >
> > > **On Domain Expertise.** We apologize for being unclear about this, and we see how this could be read as contradictory – we will clarify this in the final version. We meant that statement to apply to ChemProt and Banking77, but also note that we have results with a non-CS domain expert for Chemprot. We will overhaul these statements entirely in the final version.

---

> > > > ### Author Response · Authors · 2024-08-27
> > > >
> > > > Dear Reviewers,
> > > >
> > > > As the discussion period closes soon, we would like to thank you again for your feedback, questions, and suggestions! We believe that our new experiments in response to your questions and having incorporated your feedback have improved our paper! If you have additional questions before the end of the discussion period, we would love to answer them!
> > > >
> > > > The Authors

---

> > > > ### Comment · Reviewer_2MXd · 2024-08-27
> > > > **Thanks for your response**
> > > >
> > > > I appreciate your effort in addressing my concerns. However, I think the current draft definitely does not meet the conference standard and it may take some extensive effort to integrate the updates into your paper. As I'm evaluating based on the current version and am not able to check what the updated draft looks like, I can only raise my score to 5 but not higher.

---

> > > > > ### Author Response · Authors · 2024-08-30
> > > > >
> > > > > Thank you for your response and for raising your score. We appreciate your helpful suggestions and have incorporated them in our internal manuscript. While we cannot share the updated manuscript as part of this response, **we'd like to clarify that these updates were straightforward and did not fundamentally change our main claims.** These additions primarily involve textual clarifications and additional results which **reinforce/supplement our original claims rather than altering the paper's core takeaways.**
> > > > >
> > > > > Here's a summary of the key changes.
> > > > >
> > > > > 1. **Clarified scope and additional NER experiments.** We've explicitly stated that our primary focus is on text classification tasks, which are more common in practice and were the basis of our original findings. As suggested, we've also included supplementary results for NER datasets and pointed to this also being an important area of future work.
> > > > > 2. **In-domain pre-trained models.** We've added experiments using domain-specific models like Legal BERT and SciBERT, which led us to the same conclusions as using the RoBERTA-base model that we were using before, which enhances the robustness of our results.
> > > > > 3. **Comprehensive metrics.** We've included additional F1 scores (micro, macro, weighted) for all datasets in the appendix, providing a more complete picture of performance across imbalanced classes. The inclusion of these metrics does not change any of our conclusions, and in fact shows that our results are robust to changes in the metric.
> > > > > 4. **Enhanced LF details.** We've added a dedicated section in the main paper discussing LF design, development time, and impact on performance, with further details in the appendix.
> > > > > 5. **Refined limitations section.** We've clarified our statements about domain expertise to avoid any inconsistencies.
> > > > >
> > > > > We appreciate your time and consideration, and we're happy to provide any further clarification if needed.

---

### Official Review · Reviewer_YJ8S · 2024-07-14

**Rating:** 9
**Confidence:** 5
**Correctness:** yes
**Clarity:** yes

**Review:**

Clarity
- The paper is clear and well-organized, though some details can be more elaborated in the paper or the supplementary materials.

Originality
- The work is original in its development of BOX WRENCH, a new benchmark for evaluating weak supervision systems. This benchmark addresses limitations of previous benchmarks by incorporating higher class cardinality, class imbalance, and domain expertise requirements, thus better reflecting real-world applications.

Significance
- The significance of this work is substantial. It provides a valuable empirical study that enhances the understanding of the practicality and usability of weak supervision (WS) methods. The provided labeling function and datasets can be used for practitioners in this field.

**Strengths:**

See above.

**Additional Feedback:**

n/a

**Documentation:**

yes

**Limitations:**

yes

**Opportunities For Improvement:**

Overall, I am positive about this paper, but I have some questions that I would like to discuss with the authors:

Details of Labeling Functions
- This paper explores classification under high cardinality, which is less studied in WS scenarios. However, I am concerned about the labeling effort required in this setting. In Lines 167-169, the authors mention: "We manually inspected labeled examples in the development set, identified patterns for each class, and then created multiple keyword, dictionary-based, and regular expression-based LFs for each class." I would appreciate more details on the LFs and the time cost for creating them compared to manual annotation of each example. Providing further explanations would be beneficial.

Corrected LF on ChemProt
- In Section 4.2, the authors state that they refined the LF from previous papers (Yu et al., 2021), but the details are lacking. It would be helpful to provide more information on this refinement process, as well as on the LFs for other datasets introduced in this paper.

Comparison with Using LLMs
- Given that LLMs have demonstrated strong zero-shot/few-shot capabilities, I wonder if the tasks presented could be addressed by simply prompting LLMs with zero or a few demonstrations. Additionally, using LLMs to assist in LF creation could potentially enhance the LF creation pipelines. I would like to know the authors' opinion on this approach.

Additional Baselines
- The authors have provided empirical comparisons with representative baselines, but many other baselines mentioned in the WRENCH benchmark (see https://github.com/JieyuZ2/wrench?tab=readme-ov-file#-available-models) were not included. Including some of these additional baselines would be beneficial. Nonetheless, the current comparisons seem adequate for this paper, and I encourage the authors to expand these comparisons gradually.

Learning with Clean & Noisy Labels
- In Section 3.2, the authors mention the CFT method used by Zhu et al. (2023). However, I am unsure if this is the best way to combine clean and weak labels. Could using this method potentially undermine the capabilities of WS approaches with clean labels?

Clarification in Appendix E
- In Appendix E, what do "stratified" and "unstratified" mean? Providing a clear definition would be helpful.

Some papers that might be relevant to this study (about using indirect supervision to create training data for fine-grained classification):
- Zhang et al. "Creating Training Sets via Weak Indirect Supervision." ICLR 2022.

**Relation To Prior Work:**

mostly

**Summary And Contributions:**

This work introduces BoxWRENCH, a new benchmark designed to more closely resemble real-world WS scenarios. This benchmark includes challenges such as higher class cardinality, class imbalance, significant domain expertise requirements, and linguistic variations in parallel corpora. The authors also refine existing benchmark LFs through a rigorous procedure that aims to replicate real-world conditions.

---

> ### Author Rebuttal · Authors · 2024-08-17
>
> We are grateful for your insights, which have helped us to further refine our work.
>
> **On improved LFs for Chemprot.** Due to limited space, we did not fully describe all the details. We have shared the exact notebook we used to develop the LFs in the repo. We acknowledge that it could be useful to study each LF performance. We have included the procedure for improvement and instructions on further analysis of LFs in the common response.
>
> **On time spent on LF development.** We agree that a comparison between the time spent developing LFs and manual labeling is an important consideration for weak supervision, in general. Note that this comparison is a function of dataset size, as manual labeling time scales with the number of points, while weak supervision scales with the number of labeling functions (which is not dependent on the total amount of data). **As a result, weak supervision will win out for large dataset sizes, where indeed this technique is known to shine.** For the datasets we considered, the time required is approximately a few hours. We will include this detailed information in our new results, which feature newly designed LFs created by a neuroscience student specifically for the ChemProt dataset.
>
> **On comparison with/using LLMs.** Thank you for the kind suggestion! Since submission, we have already conducted  experiments exploring this point. Specifically, we prompted GPT-4o as if it was a domain expert (e.g. “you are an expert in legal document classification and label function writing”) and asked it to create keywords-based LFs based upon a development set. The full list of our prompts is included in our [codebase](https://github.com/jeffreywpli/stronger-than-you-think/tree/main/end_model_training/lable_function). While the coverage of the GPT-4o-generated label functions is higher, the precision of the LFs is generally lower (see Table1). However, we note that the generated labeling functions (LFs) are acceptable, especially considering the reduction in time cost and the potential for full automation. Future work could involve analyzing a multi-agent scenario that automates and reinforces the LF generation process. Additionally, we conducted an analysis using LLMs with zero-shot prompting. We sampled 250 data points from our test set, utilized API calls with GPT-4o-2024-08-06 for each example, and recorded the accuracy. We attached the cross-over points graph with this baseline (See Figure 1). The LLM baseline performed poorly in tasks requiring domain-specific knowledge, such as Claude9 and ChemProt. It performed well on Amazon31, possibly because the test set may be included in the public [amazon review datasets](https://amazon-reviews-2023.github.io), which GPT may have been trained on.
>
>
> ### **Table 1: Coverage and Precision**
> | Dataset    | Our Coverage | Our Precision | LLM Coverage | LLM Precision |
> |------------|--------------|---------------|--------------|---------------|
> | Amazon31   | 0.63         | 0.70          | 0.99         | 0.26          |
> | Banking77  | 0.58         | 0.81          | 0.98         | 0.62          |
> | Chemprot   | 0.82         | 0.70          | 0.96         | 0.56          |
> | Claude9    | 1            | 0.91          | 0.91         | 0.15          |
>
> **On additional baselines.** We agree that incorporating additional weak supervision methods would be beneficial. Some methods in the WRENCH benchmark, such as logistic regression and MLP, are simpler and exhibit less competitive performance compared to BERT. We are currently exploring other methods in WRENCH, such as COSINE and ARS2, which are more computationally intensive due to their 10-20 hyperparameters. We aim to obtain and present the results during the discussion period.
>
> **On better methods for learning with clean & noisy labels.**
> - *“Could using this method potentially undermine the capabilities of WS approaches with clean labels?”*\
> This is a valid concern. Indeed there are potentially many ways to combine learning from WS and a handful of clean labels. We chose CFT based on the results from Zhu et al. (2023), who found it to be most effective compared to some other baselines they tried such as mixing all the labels data together. Overall, on our tasks, we also find that CFT almost always increases performance compared to just WS alone. Overall, developing better strategies for learning from weak + clean labels is an interesting direction for future work and could potentially increase the cross-over points further.
>
> **On clarification in Appendix E.** Sorry for the confusion here! These terms refer to how we sample the small amount of clean labels in our setups. Unstratified is a uniform sample over all the labels whereas stratified refers to sampling an equal number of random examples from each class.

---

> > ### Comment · Reviewer_YJ8S · 2024-08-18
> >
> > Thanks for your reply! I will re-evaluate the paper when more results are available.

---

> > ### Author Rebuttal · Authors · 2024-08-25
> >
> > Thank you for your reply. We provide more LF improvement details ([GitHub](https://github.com/jeffreywpli/stronger-than-you-think/tree/main/end_model_training/lable_function/chemprot)), and the results of COSINE and ARS2, which are other end models available in Wrench (See attached PDF and the updated main rebuttal).

---

> > > ### Comment · Reviewer_YJ8S · 2024-08-26
> > >
> > > Thanks for the additional results. I don't have other questions and have updated my score.

---

### Official Review · Reviewer_f8Lj · 2024-07-25
**Nontrivial Weak Supervision Benchmark**

**Rating:** 7
**Confidence:** 4
**Clarity:** This paper is relatively easy to follow.

**Review:**

I think this paper is well-motivated and the overall quality of the work is high. I believe the authors have put considerable effort into task selection and weak labelers development for this benchmark. Comparing to its precursor Wrench [1], BoxWrench has a more realistic selection of tasks as the some of the prior datasets are constructed for weak supervision. This paper also included sufficient experiments to evaluate a couple classic label models.

Minor Improvement Suggestion:
1) Following prior benchmark Wrench, this paper could definitely benefit from benchmarking a wider range of weak supervision methods that rely on weak labeler votes.

Overall, I believe BoxWrench is a nontrivial contribution given its prior work Wrench. Thus I recommend acceptance.

[1] Zhang, Jieyu, et al. "WRENCH: A comprehensive benchmark for weak supervision." in NeurIPS 2022.

**Strengths:**

I believe this paper is well-motivated and of high quality. The authors have clearly invested significant effort into selecting tasks and developing weak labelers for this benchmark. The experiments are also documentary and beneficial for the broader community.

**Additional Feedback:**

I have read the author's response and I am satisfied with the rebuttal. I will keep my decision of acceptance.

**Correctness:**

To my eyes, the data collection, labeling functions development process is correct and I believe the experiment setup is also correct.

**Documentation:**

There are more detailed descriptions in supplementary/on github.

**Limitations:**

The authors have properly addressed the limitation.

**Opportunities For Improvement:**

Following prior benchmark Wrench, this paper could definitely benefit from benchmarking a wider range of weak supervision methods that rely on weak labeler votes.

**Relation To Prior Work:**

I believe this paper has addressed related work in the weak supervision benchmarking space.

**Summary And Contributions:**

This paper proposes a new weak supervision benchmark (BoxWrench) to evaluate weak supervision methods on more realistic tasks. According to the authors, the precursor of BoxWrench, Wrench benchmark is overly easy thus drive the need to develop a more realistic weak supervision benchmarks. The authors incorporate Banking77, ChemProt, Claude9, Massive18, Amazon31 and develop labeling functions for the datasets. The authors conduct expertiments to compare various label models under different settings and compared results with fully supervised classifiers trained with different amount of validation data to provide a "crossover" point analysis. Overall, the main contributions lie in the collection of the realistic tasks and develop the programmatic labelers associated with the tasks.

---

> ### Author Rebuttal · Authors · 2024-08-17
>
> Thank you for your time and we’re glad you appreciated the paper!
>
> **On benchmarking more WS methods.**  We acknowledge that incorporating additional weak supervision methods would be beneficial. Some methods in the WRENCH benchmark, such as logistic regression and MLP, are simpler and exhibit less competitive performance compared to BERT. We are currently exploring other methods in WRENCH, such as COSINE and ARS2, which are more computationally intensive due to their 10-20 hyperparameters. We aim to obtain and present the results during the discussion period.

---

> > ### Author Rebuttal · Authors · 2024-08-25
> >
> > Thank you again. We provide results of COSINE and ARS2, which are other end models available in Wrench (See attached PDF and the updated main rebuttal).

---

> > > ### Author Response · Authors · 2024-08-27
> > >
> > > Dear Reviewers,
> > >
> > > As the discussion period closes soon, we would like to thank you again for your feedback, questions, and suggestions! We believe that our new experiments in response to your questions and having incorporated your feedback have improved our paper! If you have additional questions before the end of the discussion period, we would love to answer them!
> > >
> > > The Authors

---

### Official Review · Reviewer_QsfA · 2024-08-01
**A new benchmark for weak supervision; lack in-depth discussion**

**Rating:** 5
**Confidence:** 5
**Correctness:** Yes
**Clarity:** Yes

**Review:**

This paper is generally easy to follow. However, I find it's less sound and lacks in-depth discussion of the experimental results.

I have the following concerns:

1.	The conclusions section is less informative. As a benchmark paper, I would like to defend WS with a new dataset and a series of experiments. The authors didn’t well carry out their main conclusions which may shed light on the community.

2.	Without carrying out informative takeaways, I find that ONE empirical takeaway of this paper can be rather trivial --- it is obvious that better LFs result in better WS performance.

3.	In experiments, the authors tested different LMs, but I have no idea what we can infer from this experiment series. The authors didn’t provide any discussions.

4.	The authors claim that they designed a new LF. But the technical details are too simple to understand. Moreover, the authors didn’t systematically verify the powerfulness of the newly proposed LF but only conducted one single simple experiment to support its effectiveness, which can be less persuasive.

**Strengths:**

Pros:

1.	Provides a new benchmark for weak supervision with more labels and imbalanced label distributions, which is much more realistic.

2.	The concept of a cross-over point is straightforward.

3.	The new idea of validating the transferability of WS is novel.

**Additional Feedback:**

NA

**Documentation:**

Yes

**Limitations:**

Yes

**Opportunities For Improvement:**

Add more discussion of the experimental results.

**Relation To Prior Work:**

Yes

**Summary And Contributions:**

This paper studies the weak supervision for label-efficient learning and attempts to show that WS may be stronger than is thought. In particular, the author would like to provide a new benchmark dataset with an imbalanced distribution and more classes and also verify the importance of designing better LFs. The benchmark datasets are publicly released.

---

> ### Author Rebuttal · Authors · 2024-08-17
>
> Thank you for your time and helpful suggestions!
>
> **On significance of results.** We included five very different datasets and conducted new experiments that utilized the cross-over points to show the performance of WS methods against supervised-only methods on both old datasets and new ones. Our experiments expand upon Zhu et al.’s (2023) work by looking at finer-grained proportions of the labeled data. More clean data are required for supervised-only methods to beat WS methods on all of our new datasets (ranging from hundreds to thousands) compared to the old datasets (around 50s). This is evident that these new datasets are harder and could provide more insights as WS benchmarks. In addition, we also included a novel study of the portability of LFs, using them across different languages.
>
> - *“I find that ONE empirical takeaway of this paper can be rather trivial --- it is obvious that better LFs result in better WS performance.”*\
> We agree that this statement on its own is not surprising. However, we believe it is a critical nuance about WS benchmarking that has been often ignored by the field. Most of the time, studies continue to use the same fixed LF sets from WRENCH, which Zhu et al. (2023) show are only as effective as a handful of hand-labeled examples. Further, while Zhu et al. use their results to argue that the effectiveness of WS methods is overstated, we discuss LF improvements to show that this actually reflects a failure of the benchmarks. By having low-quality LF sets, benchmarks may not be reflective of the realistic regimes where WS is useful.
>
> - *“In experiments, the authors tested different LMs, but I have no idea what we can infer from this experiment series. The authors didn’t provide any discussions.”*\
> We see this set of results as answering two questions: First, as in the original WRENCH benchmark, we wanted to see which WS label models perform best on different tasks. Second, we wanted to see if our conclusions about cross-over points change depending on which label model we use. Overall we found that the cross-over points are consistently above 25% of the validation size, and the majority of them are above 50% of the validation size, indicating the benefits of WS are not sensitive to the choice of label models.
>
> **On effectiveness of new LFs.**
> Thank you for the feedback on this aspect of the writing – we will clarify our exposition of the LF design in the final version. We reported the accuracy, precision, and coverage of the improved LFs in our paper. We also provided an analysis of LFs in our repository, along with methods for evaluating the performance of individual LFs (see the common response for details). Is there any specific analysis that you would like to see?

---

> > ### Author Response · Authors · 2024-08-27
> >
> > Dear Reviewers,
> >
> > As the discussion period closes soon, we would like to thank you again for your feedback, questions, and suggestions! We believe that our new experiments in response to your questions and having incorporated your feedback have improved our paper! If you have additional questions before the end of the discussion period, we would love to answer them!
> >
> > The Authors

---

### Author Rebuttal · Authors · 2024-08-17

We thank the reviewers for their kind comments and input. Before proceeding with in-depth responses, we highlight the strengths of our work as noted by reviewers.

- Our tasks address important limitations in previous benchmarks with higher class cardinality, class imbalance, and domain expertise requirements.
- Using cross-over points as a straightforward approach to compare WS methods and supervised methods
- Development and improvements on LF sets for new and existing tasks.

In the common response, we will discuss a common point that was raised by more than one reviewer.

**On the need for more documentation of new LF sets.** Due to limited space, we provided a necessarily compressed description of the details for  each LF.  We agree that it will be useful to allow others to study the specific LFs we proposed and have thus shared the exact notebooks we used to develop the LFs in our repo. We also created an additional section in the appendix that describes the LF improvement procedure in detail. We also attached it here for reference:\
**Procedure to improve the LFs:**
1. We first look at the coverage, conflicts ratio, and accuracy for all the existing LFs on the sampled development dataset. We identify the LFs that perform poorly and labels that need better LFs.
- *Coverage: The proportion of data points for which a label function provides a label.*
- *Conflict Ratio: The proportion of labeled data points where multiple label functions disagree.*
- *All of these statistics can be obtained from open-source Snorkel library functions, and are provided in the shared notebook in our codebase for further exploration (see [GitHub](https://github.com/jeffreywpli/stronger-than-you-think/tree/main/end_model_training/lable_function)).*
2. We look through the definition for each label, and our sampled data. Carefully examine them and observe patterns.
3. Since the original LFs are all keywords-based, we first improved the keywords themselves. For example, we include synonyms, adding space around the word to make it more accurate.
4. We incorporate positions of words in some label functions. For example, the words ‘increase’ and ‘decrease’ can appear in one sentence. With position encoded in our LFs, we can check whether the keyword we are looking for is between the two entities or around one of the entities.

---

> ### Author Rebuttal · Authors · 2024-08-25
>
> Here we provide the additional results for COSINE and ARS2 (See attached PDF). We used both of the learning methods with a RoBERTa/Legal BERT backbone on the WS-only pipeline. The other setups were kept the same. The ARS2 results follow a similar trend to our previous results, with almost identical cross-over points. COSINE performance is slightly worse on Amazon31 and Banking77, and is slightly better on Claude9, when compared with the supervised-only method.

---

### Decision · Program_Chairs · 2024-09-26

**Decision:**

Accept (Poster)

**Comment:**

This paper introduces a new benchmark, BOXWRENCH, which is designed to more accurately reflect the real-world usage of weak supervision. This benchmark features higher class cardinality and imbalance, substantial domain expertise requirements, and linguistic variations found in parallel corpora. The authors also refine existing benchmark labeling functions through a rigorous procedure that aims to replicate real-world conditions.

This paper is well-motivated and the overall contribution is high.

This paper finally receives the scores of 9, 7, 5, 5. The authors have provided a rebuttal, which addressed most of the concerns raised by the reviewers. To further improve the paper, I recommend that the authors provide more informative takeaways and make more updates according to the discussions with the reviewers.